# A novel yeast hybrid modeling framework integrating Boolean and enzyme-constrained networks enables exploration of the interplay between signaling and metabolism

**Linnea Österberg**[1,2,3], **Iván Domenzain**[3,4], **Julia Münch**[1,2], **Jens Nielsen**[3,4,5], **Stefan Hohmann**[3], **Marija Cvijovic**[1,2]*

**1** Department of Mathematical Sciences, University of Gothenburg, Gothenburg, Sweden, **2** Department of Mathematical Sciences, Chalmers University of Technology, Gothenburg, Sweden, **3** Department of Biology and Biological Engineering, Chalmers University of Technology, Gothenburg, Sweden, **4** Novo Nordisk Foundation Center for Biosustainability, Chalmers University of Technology, Gothenburg, Sweden, **5** BioInnovation Institute, Copenhagen, Denmark

* marija.cvijovic@chalmers.se

**Citation:** Österberg L, Domenzain I, Münch J, Nielsen J, Hohmann S, Cvijovic M (2021) A novel yeast hybrid modeling framework integrating Boolean and enzyme-constrained networks enables exploration of the interplay between signaling and metabolism. PLoS Comput Biol 17(4): e1008891. https://doi.org/10.1371/journal.pcbi.1008891

**Data Availability Statement:** All relevant data are within the manuscript and its Supporting Information files. The model, code, and datasets

## Abstract

The interplay between nutrient-induced signaling and metabolism plays an important role in maintaining homeostasis and its malfunction has been implicated in many different human diseases such as obesity, type 2 diabetes, cancer, and neurological disorders. Therefore, unraveling the role of nutrients as signaling molecules and metabolites together with their interconnectivity may provide a deeper understanding of how these conditions occur. Both signaling and metabolism have been extensively studied using various systems biology approaches. However, they are mainly studied individually and in addition, current models lack both the complexity of the dynamics and the effects of the crosstalk in the signaling system. To gain a better understanding of the interconnectivity between nutrient signaling and metabolism in yeast cells, we developed a hybrid model, combining a Boolean module, describing the main pathways of glucose and nitrogen signaling, and an enzyme-constrained model accounting for the central carbon metabolism of *Saccharomyces cerevisiae*, using a regulatory network as a link. The resulting hybrid model was able to capture a diverse utalization of isoenzymes and to our knowledge outperforms constraint-based models in the prediction of individual enzymes for both respiratory and mixed metabolism. The model showed that during fermentation, enzyme utilization has a major contribution in governing protein allocation, while in low glucose conditions robustness and control are prioritized. In addition, the model was capable of reproducing the regulatory effects that are associated with the Crabtree effect and glucose repression, as well as regulatory effects associated with lifespan increase during caloric restriction. Overall, we show that our hybrid model provides a comprehensive framework for the study of the non-trivial effects of the interplay between signaling and metabolism, suggesting connections between the Snf1 signaling pathways and processes that have been related to chronological lifespan of yeast cells.

used for this study can be found in the GitHub repository YeastHybridModelingFramework https://github.com/cvijoviclab/YeastHybridModelingFramework.

**Funding:** This work was supported by the Swedish Research Council (VR2016-03744) to SH, Swedish Foundation for Strategic Research (Grant Nr. FFL15-0238) to MC and European Union's Horizon 2020 research and innovation program, project CHASSY (grant agreement 720824) to ID. Part of this work was funded by the Novo Nordisk Foundation (grant no. NNF10CC1016517) and the Knut and Alice Wallenberg Foundation to JN. The funders had no role in study design, data collection and analysis, decision to publish, or preparation of the manuscript.

**Competing interests:** The authors have declared that no competing interests exist.

## Author summary

Elucidating the complex relationship between nutrient-induced signaling and metabolism represents a key in understanding the onset of many different human diseases like obesity, type 3 diabetes, cancer, and many neurological disorders. In this work we proposed a hybrid modeling approach, combining Boolean representation of signaling pathways, like Snf1, TORC1, and PKA with the enzyme constrained model of metabolism linking them via the regulatory network. This allowed us to improve individual model predictions and elucidate how single components in the dynamic signaling layer affect steady-state metabolism. The model has been tested under respiration and fermentation, revealing novel connections and further reproducing the regulatory effects that are associated with the Crabtree effect and glucose repression. Finally, we show a connection between Snf1 signaling and chronological lifespan.

## Introduction

Biological systems are of complex nature comprising numerous dynamical processes and networks on different functional, spatial and temporal levels, while being highly interconnected [1]. The field of systems biology faces the great challenge of elucidating how these interconnected systems work both separately and together to prime organisms for survival. One such phenomenon is the cells' ability to sense and respond to environmental conditions such as nutrient availability. To coordinate cellular metabolism and strategize, the cell needs an exact perception of the dynamics of intra- and extracellular metabolites [2]. Simultaneously, nutrient-induced signaling plays a pivotal role in numerous human diseases like obesity, type 2 diabetes, cancer and aging [3–6]. Therefore, unraveling the role of nutrients as signaling molecules and metabolites as well as their interconnectivity may provide a deeper understanding of how these conditions occur.

Yeast has long been used as a model organism for studying nutrient-induced signaling [7]. Two major classes of nutrients include carbon and nitrogen. Carbon-induced signaling acts mainly through the PKA and SNF1 pathway while nitrogen-induced signaling acts through the mTOR pathway. The PKA pathway plays a major role in regulating growth by inducing ribosome biogenesis genes and inhibiting stress response genes [8]. The SNF1 pathway is mainly active in low glucose conditions where it promotes respiratory metabolism, glycogen accumulation, gluconeogenesis, and utilization of alternative carbon sources but it also controls cellular developmental processes such as meiosis and aging [7, 9, 10]. The strongly conserved TORC1 pathway plays a crucial role in promoting anabolic processes and cell growth in response to nitrogen availability [8]. Active TORC1 induces ribosomal protein and ribosome biogenesis gene expression [11, 12] and represses transcription of genes containing STR and PDS elements in their promoter region [11]. Even though Snf1, TORC1, and PKA pathways belong to the most well-studied pathways [2], there is still a lack of understanding both in the dynamics and the interactions leading to change in gene expression. It has been shown, that glucose signaling is related to metabolism however the nature of this relationship remains unknown [13]. Numerous crosstalk mechanisms between these pathways have been described [14], and depending on their activity, they may influence the overall effect of the signaling process and thus the interaction with the metabolism [15]. To better understand the impact of cell signaling on metabolism, a systems biology approach is often implemented [16].

Typically, Boolean models have been developed to study the crosstalk between the Snf1 pathway and the Snf3/Rgt2 pathway [17] as well as the Snf1, cAMP-PKA, and Rgt2/Snf3 pathways [15]. In mammalian cells, Boolean models have been used to evaluate the conflicting hypothesis of the regulation of the mTOR pathway [18] and to study crosstalk between mTOR and MAPK signaling pathways [19]. Since, signaling systems are not always strictly Boolean in its nature, where location, combinations of post-translational modifications as well as other interaction play a role, alternative Boolean frameworks for handling these complex interactions have been developed [15, 20]. In contrast, metabolism, also in itself a complex process, is often studied using Flux Balance Analysis (FBA), which enables prediction of biochemical reaction fluxes, cellular growth on different environments, and gene essentiality even for genome-scale metabolic models [21–23]. A major limitation of the use of GEMs together with FBA is the high variability of flux distributions for a given cellular objective [24], as FBA solves largely underdetermined linear systems through optimization methods. To overcome this problem, experimentally measured exchange fluxes (uptake of nutrients and secretion of byproducts) are incorporated as numerical constraints, however, such measurements are not readily available for a wide variety of organisms and growth conditions.

The concept of enzyme capacity constraints has been incorporated into FBA to reduce the phenotypic solution space (i.e. exclusion of flux distributions that are not biologically meaningful) and diminish its dependency on condition-dependent exchange fluxes datasets [25–30]. Notably, a method to account for enzyme constraints, genome-scale models using kinetics and omics (GECKO; Sánchez et al., 2017) has been developed. GECKO incorporates constraints on metabolic fluxes given by the maximum activity of enzymes, which are also constrained by a limited pool of protein in the cell. This method has refined predictions for growth on diverse environments, cellular response to genetic perturbations, and even predicted the Crabtree effect in *S. cerevisiae*'s metabolism, but also proven to be a helpful tool for probing protein allocation and enabled the integration of condition-dependent absolute proteomics data into metabolic networks [28, 30].

Following the holistic view of systems biology, hybrid models allow us to take the next step and combine different formalisms to study the interconnectivity and crosstalk spanning different scales and/or systems. For example, to quantify the contribution of the regulatory constraints of an *Escherichia coli* genome-scale model, a steady-state regulatory flux balance analysis (SR-FBA) has been developed [31]. Furthermore, the diauxic shift in *S. cerevisiae* has been studied by the CoRegFlux workflow, integrating metabolic models and gene regulatory networks [32]. To bypass the need for kinetic parameters, a FlexFlux tool has been developed where metabolic flux analyses using FBA have been constrained with steady-state values resulting from the regulatory network [33]. This strategy has also been used in a hybrid model of *Mycobacterium tuberculosis* where the gene regulatory network was used to constrain the metabolic model to study the adaptation to the intra-host hypoxic environment [34]. However, to further study the impact of signaling on metabolism, the complexity of the signaling systems itself and the crosstalk between interacting pathways need to be represented coherently.

To better understand the complex relationship between metabolism and signaling pathways, we created a hybrid model consisting of a Boolean module integrating the PKA, TORC1, and the Snf1 pathways as well as the known crosstalk between, together with an enzyme-constrained module of *S. cerevisiae*'s central carbon and energy metabolism (Fig 1). The backbone of the presented model is a framework for utilizing the complex Boolean representation of large-scale signaling systems to further constrain an enzyme-constrained model (ecModel) of the central carbon metabolism. With the glucose level as input, transcription factor activities, resulting from the Boolean module are mapped to a regulatory network. The bounds of the solution space, calculated using enzyme usage variability analysis on the genes affected by the

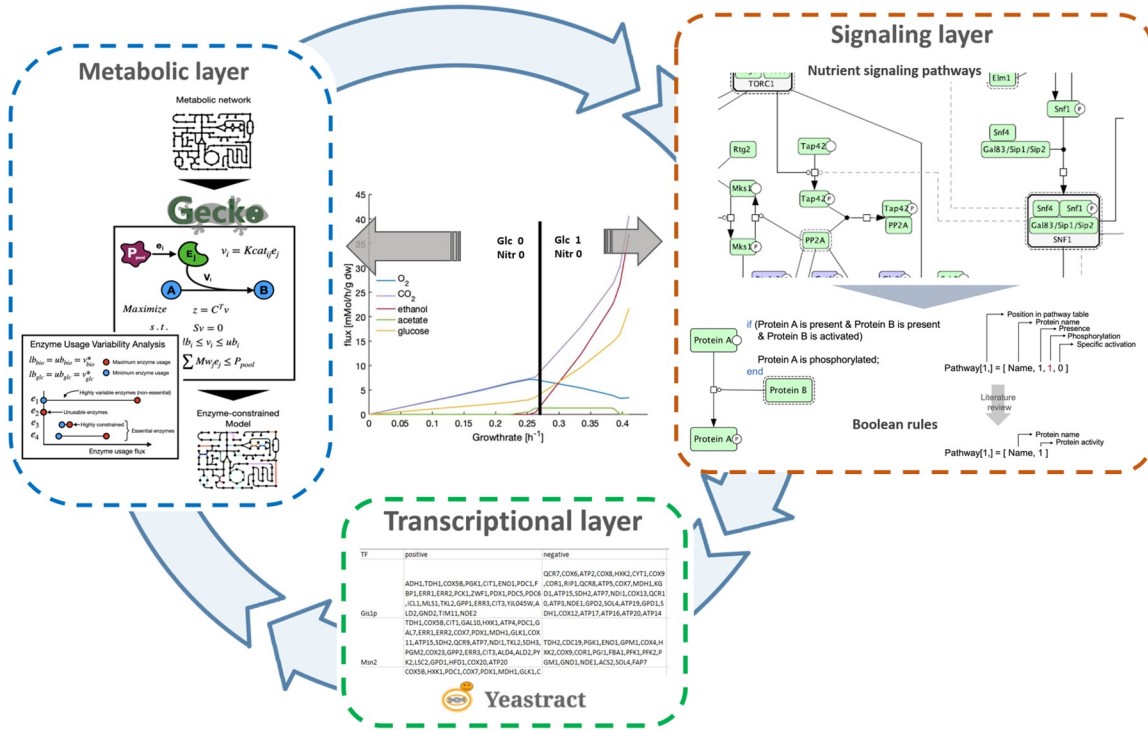

**Fig 1. Schematic representation of the hybrid model.** The hybrid model consists of a vector-based Boolean module of nutrient signaling and an enzyme constrained module of the central carbon metabolism. The Boolean module is a dynamic module including Snf1, PKA, and TORC1 pathway as well as crosstalk between them. The dynamic module reaches a steady-state and the activity of the transcription factors acts as input in a regulatory network constraining the enzyme constraint model of the central carbon metabolism. The solution is used to determine the activity of the Boolean input.

transcription factors, are altered depending on up- or down- regulation and used to constrain an ecModel of the central carbon metabolism (for details see Method section). The predictions of protein allocation, at the individual enzyme level for respiratory and fermentative conditions, are improved by the incorporation of the regulatory layer into the hybrid model, in comparison with its pure enzyme-constrained counterpart. Moreover, the predicted enzyme usage profiles display a diversified utilization of isoenzymes, which is supported by proteomics data, but previous constraint-based methods have failed to capture. Additionally, the proposed hybrid model is capable of reproducing the regulatory effects that are associated with the Crabtree effect and glucose repression and have further showed a connection between glucose signaling and chronological lifespan by the regulation of NDE and NDI usage in respiring conditions. Finally, the model showed that during fermentation, enzyme utilization is the more important factor governing protein allocation, while in low glucose conditions robustness and control are prioritized.

## Results

### Implemented Boolean signaling network reproduces the general dynamics caused by glucose and nitrogen addition to starved cells

To verify the constructed Boolean model of nutrient-induced signaling pathways (Fig 2), cells were simulated from nitrogen- and glucose-starved conditions to nutrient-rich conditions. We also simulated the model from nutrient-rich conditions to nutrient depletion. The

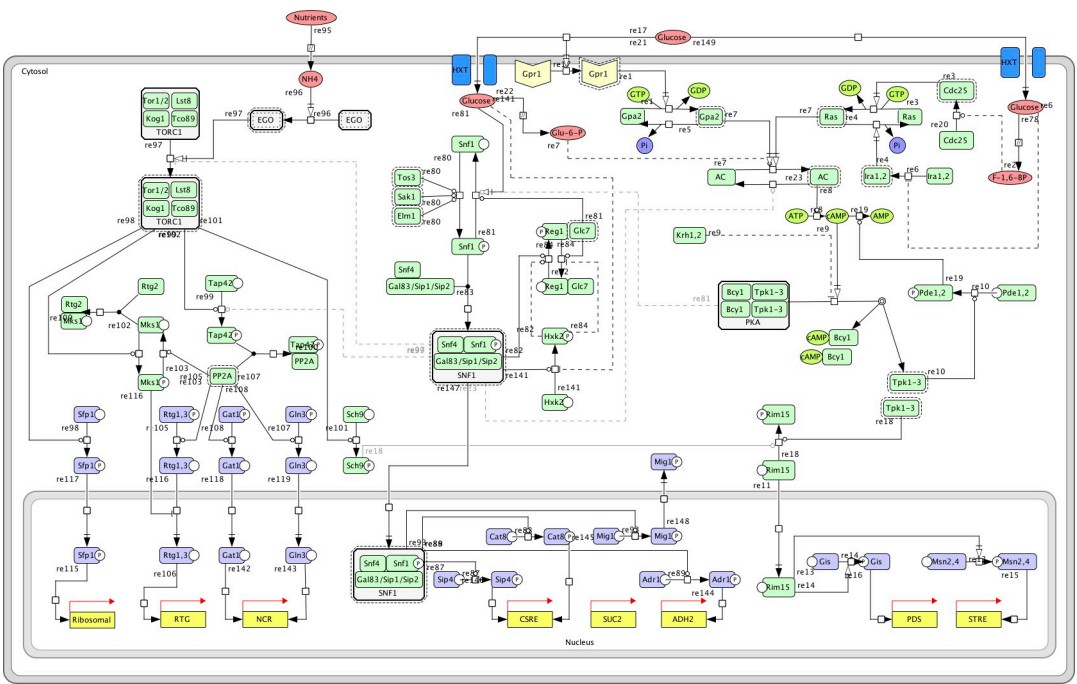

**Fig 2. The Boolean module is a dynamic module including Snf1, PKA, and TOR pathway as well as crosstalk between them.** Crosstalk events between the pathways are depicted in grey. Unknown mechanisms are represented by dashed lines.

simulated results were compared to the literature concerning both dynamics and steady-state outcome. An in-depth literature review of the known mechanisms and its implementation and interpretation in the model as well as a graphical representation of the simulated dynamics are available in S1 Text, S1 Fig, and S1 Table. The PKA pathway was activated upon glucose abundance via the small G proteins Ras and Gpa2. These proteins, in turn, activated the adenylate cyclase (AC) that induced the processes leading to the activation of the catalytic subunit of PKA. Active PKA phosphorylated and therefore inactivated Rim15, thus the transcription factors Gis1, Msn2, and Msn4 became inactive. Our result pinpointed PKA as the main regulator of Rim15 (for details see S1 Text), while previous experimental studies showed that Sch9 is the major regulator of Rim15 [35]. Further, simulations from high to low nutrient conditions are in agreement with the literature on dynamics and steady-states (S1 Fig). When glucose is depleted Ira becomes active and sequentially Cdc25 gets inactivated which results in Ras inactivation. Simultaneously Gpr1 gets inactivated, turns off Gpa2 relieving the inhibitory effect on Krh activity. This inactivates AC and in turn PKA. Pde and Rim15 get dephosphorylated and Rim15 can phosphorylate Gis1 and Msn2/4.

The SNF1 pathway is active when glucose is limited, while the addition of glucose causes Snf1 inactivation resulting in the activation of the transcriptional repressor Mig1 and the deactivation of Adr1, Cat8, and Sip4. However, the inactivation of Adr1 happened before Snf1 inactivation. This is due to the implemented crosstalk with the PKA pathway, where activated PKA inhibits Adr1 activity [36]. This crosstalk has a similar effect on the dynamics of Adr1 activation in simulations from high to low nutrient conditions. Snf1 is phosphorylated by the upstream kinases when glucose is depleted and then phosphorylates Mig1, Cat8, Sip4, Adr1, and Reg1. Cat8 and Sip1 become active while Adr1 activation occurs two iterations later due to the crosstalk implementation between the PKA and SNF1 pathway (for details see S1 Text).

First, when PKA gets inactive the inhibitory effect it has on Adr1 releases. Mig1 gets inactivated and the phosphorylation of Reg1 activates Glc7.

Nutrient availability activates the TOR complex 1 which in turn phosphorylates Sch9 and Sfp1 resulting in the repression of Rim15 phosphorylation and the expression of ribosomal genes respectively. No change was observed in the activity of PP2A-regulated transcription factors Rtg1, Rtg3, Gat2, and Gln2. However, during the 8[th] iteration, PP2A was active. In addition, Sch9 was not the main regulator of Rim15 activity in our simulations since PKA was activated before Sch9 and acted independently to regulate Rim15, either due to a gap in the model or a lack of complexity in our understanding of the signaling system (S1 Text). When glucose is depleted the EGO complex loses activity which transmits to the TORC1 complex and in turn to Sch9 and Sfp1.

## The Boolean model reveals interconnectivity and knowledge gaps in nutrient signaling pathways

To further investigate the impact of nutrient conditions on the crosstalk between pathways in the Boolean model, knockouts of main components of each pathway (Snf1, Reg1, Tpk1-3, and Tor1,2) were simulated and compared to the wildtype in glc|nitr = 1|1 and glc|nitr = 0|0 (S2 Fig). In nutrient-depleted conditions, only the Snf1 knockout had a significant impact. In the Snf1 pathway, Snf1 knockout affected all downstream targets leading to a transcription factor activity pattern that is usually observed in wildtype strains when glucose is available [7]. It has been previously described that the phenotype of Snf1 mutants resembles the phenotype observed when the cAMP/PKA pathway is over-activated [37]. Although activation of the adenylate cyclase (AC) could be observed in the simulated knockout, PKA and the downstream targets were inactive due to the activity of the Krh proteins that inhibit PKA if no glucose is present in the Boolean model (S1 Text). The Snf1 mutant showed defects in the TOR pathway upon glucose depletion leading to the activation of the PP2A phosphatase. The resulting activation of NCR and RTG genes and deactivation of ribosomal genes correspond to the phenotype one would expect if glucose but not nitrogen is available [38] thus stressing the role of Snf1 in imparting the glucose state to the other nutrient-signaling pathways.

Under high nutrient availability, the Reg1 knockout showed almost the same effect on the SNF1 and TORC1 pathway as nutrient depletion. Only Adr1 activity was not affected which opposes the observations by Dombek and colleagues [39], that described constitutive ADH2 expression in Reg1 mutant cells (S1 Text).

An almost similar effect on the SNF1 and TORC1 pathways could be observed when Tpk1-3 knockout was simulated. This redundant effect was expected since impaired PKA activity was described to be associated with increased SNF1 activity[40]. Nevertheless, PKA knockout additionally induced Adr1 activation when SNF1-mediated activation could no longer be inhibited by PKA. The PKA knockout simulation showed strong effects on all three simulated pathways and may explain why strains lacking all three Tpk isoenzymes are inviable [41].

The effects of Tor1 and 2 knockouts only affected the TORC1 signaling pathway. The simulated phenotype equaled the phenotype that is expected upon nitrogen depletion and glucose abundance and was therefore similar to the phenotype observed when simulating the Snf1 knockout in nutrient-starved cells. Besides, experimental observations revealed that impairing Tor1 and 2 function results in growth arrest in the early G1 phase of the cell cycle, as well as inhibition of translation initiation which are characteristics of nutrient, depleted cells entering stationary-phase [42]. The fact that inactivation of TORC1 results in the inactivation of Sfp1 that regulates the expression of genes required for ribosomal biogenesis could be an indicator of this observation; however other TORC1-associated signaling mechanisms inducing translation initiation may likely be involved [42].

## The hybrid model improves protein allocation predictions by showing a diversified use of isoenzymes

To verify the performance of the ecModel layer, predicted exchange fluxes at increasing dilution rates on glucose-limited conditions were compared against experimental data [43] (S3 Fig), predictions showed a median relative error of 9.82% in the whole range of dilution rates from 0 to 0.4 $h^{-1}$, spanning both respiratory and fermentative metabolic regimes. The hybrid model, including regulation, was further compared with the ecModel in its ability to compute protein demands by comparing the predicted enzyme usages to protein abundance data from the literature, in both respiratory and fermentative conditions [44, 45]. Analysis of results revealed that, in respiration, 40.83% of the proteins in the model are predicted in the same order of magnitude as their experimental values, and 31.66% are predicted with an error between one and two orders of magnitude, yielding an average absolute $\log_{10}$ fold-change between predictions and measurements of 1.55. For the fermentative condition, 65.51% of the proteins are predicted within the same order of magnitude as their experimental measurements, showing an average absolute $\log_{10}$ fold-change of 2.32 (S1 Data and S2 Text). Furthermore, two-sample Kolmogorov-Smirnov tests did not show statistically significant differences between the hybrid model predictions and the available proteomics datasets.

Pathway enrichment analysis of the proteins miss-predicted by more than one order of magnitude by the hybrid model was performed using a hypergeometric distribution test and the Holm-Bonferroni correction method for multiple testing. Results showed that the super-pathway of glucose fermentation was significantly enriched for underpredicted proteins on both respiratory and fermentative conditions (p-value of $1.39 \times 10^{-7}$ and $7 \times 10^{-5}$, respectively); additionally, TCA and glyoxylate cycles showed significant enrichment for underpredicted proteins uniquely in fermentation (p-values of $3 \times 10^{-2}$). On the other hand, the super-pathways of aerobic fermentation and electron transport chain were significantly enriched for overpredicted proteins in the fermentative condition (p-value = $2.85 \times 10^{-23}$). The pentose phosphate pathway and glucose-6-phosphate biosynthesis showed significant enrichment for underpredicted proteins just in the respiratory condition (p-values of $2.86 \times 10^{-4}$ and $1.95 \times 10^{-2}$, respectively). A detailed comparison between the model predictions and in-depth results from the protein predictions are available in S1 Data, S2 Table, and S2 Text.

Comparison with the pure enzyme-constrained model showed that, by adding the regulation layer, prediction of protein demands are improved by more than one order of magnitude, on average, as the aforementioned $\log_{10}$-transformed ratio is reduced from 2.62 to 1.55, in respiration, and from 3.56 to 2.32 for fermentation. This large improvement is predominantly resulting from the utilization of more than one isoform for some reactions in the hybrid model in contrast to a pure ecModel, in which just the most efficient enzyme for a given reaction is used, due to its reliance on optimality principles.

Utilization of isoenzymes was assessed by comparing predicted non-zero enzyme usages, for different isoforms in a given metabolic reaction, to their presence in the datasets for both conditions, returning confusion matrices for the ecModel and hybrid model in each condition (S1 Data). Fig 3 provides a detailed comparison of isoenzymes presence in unregulated and regulated model predictions and proteomics datasets. Predictive performance was then evaluated by computing sensitivity, specificity, precision, accuracy and the Fowlkes-Mallow index, which takes into account all the pair of points in which two clusters of data agree or disagree, approaching the value of one for highly similar clusters [46]. Overall, these metrics revealed that the hybrid model outperforms the ecModel in its ability to predict utilization of expressed isoenzymes in both respiration and fermentation conditions. Further details on predictive performance assessment are shown in Fig 3B.

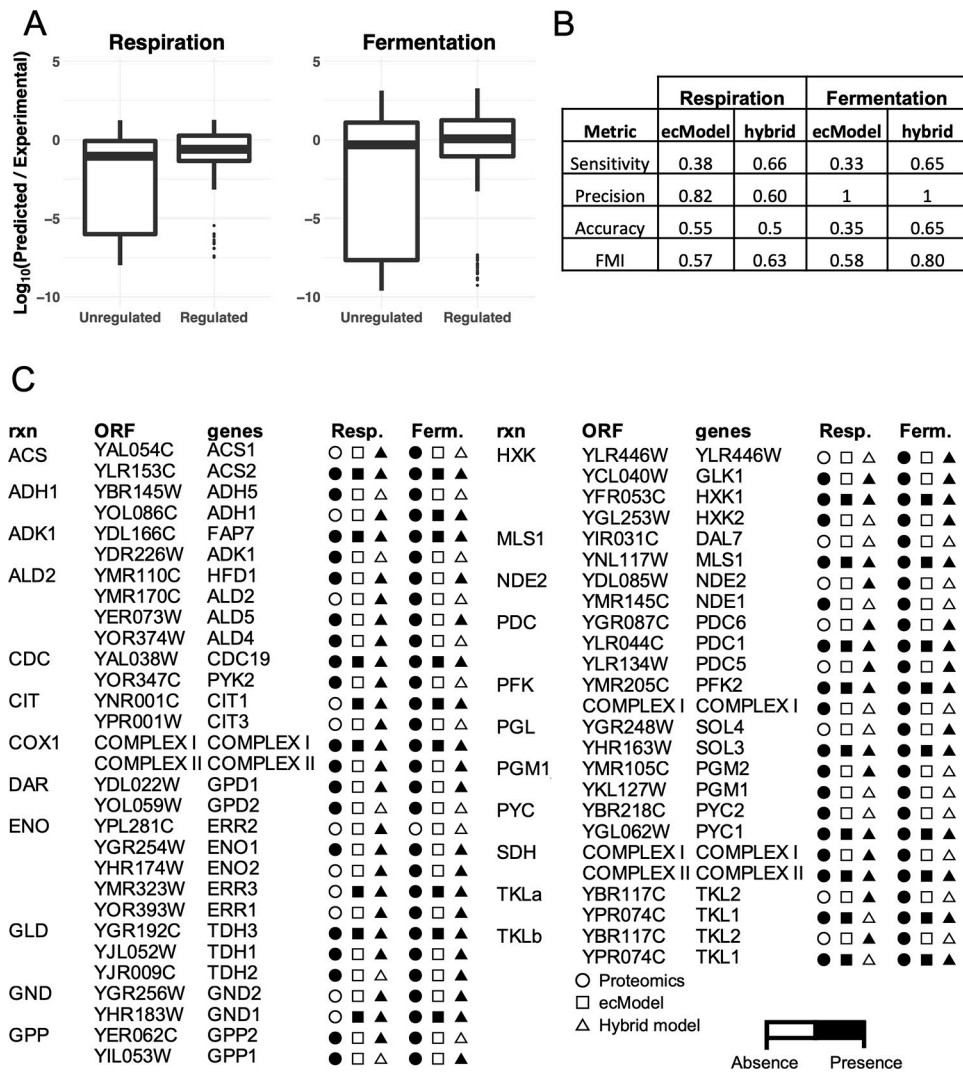

**Fig 3.** (A) Absolute log₁₀-transformed ratio between predicted and measured protein abundance values in respiration and fermentation for the purely enzyme-constrained and hybrid models. (B) Evaluation of isoenzymes utilization predictions, comparing the ecModel and hybrid model on respiratory and fermentative conditions against experimental data on protein expression (absence/presence). FMI—Fowlkes-Mallows index. (C) Comparison of individual isoenzymes utilization between models' predictions and experimental data. Color indicates presence or absence of a given protein in the predictions of the ecModel, hybrid model and experimental data on protein expression.

## The hybrid modeling framework reveals a connection between regulation and chronological aging as well as fundamental strategies of enzyme utilization

To better understand which pathways and reactions are most affected by regulation, the metabolic flux distributions predicted by the hybrid model and the ecModel were compared. Larger flux differences arose for respiratory conditions, in which the average relative change in flux was 1.85 in contrast to 0.46 in fermentation (S2 Data), this result is heavily influenced by the amount of totally activated or deactivated fluxes by the hybrid model, 57 for respiration and 29 for fermentation (Fig 4 and S2 Data). In the ecModels formalism reversible metabolic reactions are split, creating separate reactions for the forward and backward fluxes, thus distributions of

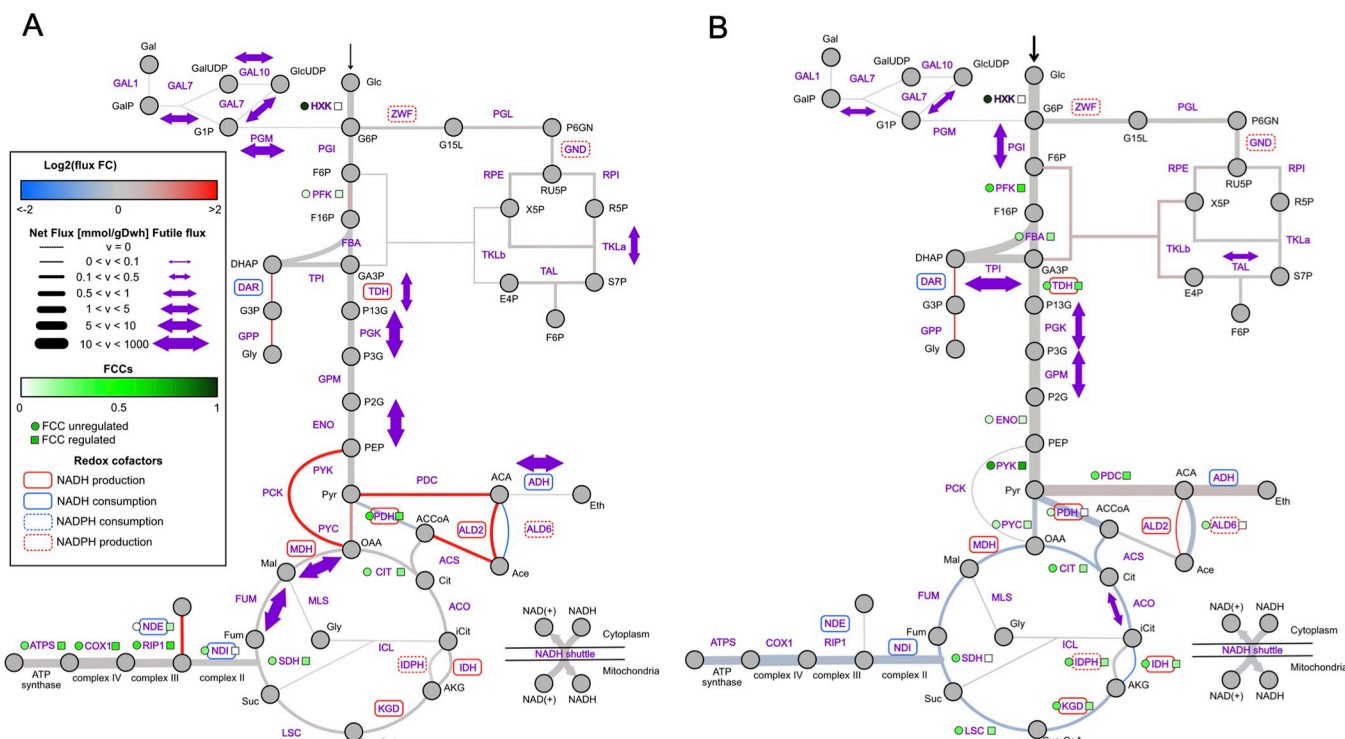

**Fig 4.** The core reactions in the metabolism under (A) respiration and (B) fermentation are shown. The fluxes are represented by the width of the connectors where dotted lines represent zero flux. The color of the connectors represents the change in flux from the unregulated ecModel compared to the regulated hybrid model. The FCCs are represented in the figure where the unregulated case is depicted by circles and compared to the regulated case depicted in squares.

net metabolic fluxes were also obtained and compared among models and conditions. As some enzymes are upregulated by the hybrid model even to levels that exceed the flux capacity of certain pathways (for a fixed growth rate), futile fluxes are expected to arise across the metabolic network.

Increased exchange fluxes for glucose, oxygen, and acetate were observed in the respiration phase (S2 Data). Additionally, an increase in the overall flux through the pentose phosphate pathway as well as an induced use of NDE instead of only NDI, allowing for the utilization of cytosolic NADH to reduce oxygen demands in the oxidative phosphorylation pathway was detected and has previously been associated with chronological aging [47]. Increased flux on PCK, PDC, ALD2, and ACS, around the pyruvate branching point, led to an overall increased flux through the TCA cycle (Fig 4 and S2 Data). To balance the increased production of AMP by ACS the ADK reaction is also upregulated. Futile fluxes are induced by regulation in galactose metabolism (GAL7 and GAL10), lower glycolysis (TDH, PGK, and ENO), TCA cycle (FUM and MDH), as well as TKLa and PGM in the pentose phosphate pathway and ADH (Fig 4 and S2 Data) in respiration.

In the fermentation state, futile fluxes also occur in galactose metabolism (GAL7) as well as glycolysis (PGI, PGK, TPI, and GPM), TAL1 in the pentose phosphate pathway, and ACO in the TCA cycle (S4 Fig and S2 Data). Down-regulation of oxidative metabolism increased uptake of glucose, and increased flux through glycolysis was observed, which is consistent with the changes that have been attributed to glucose-induced repression during the long term Crabtree effect [48] (Fig 4).

The control exerted by each enzyme on the global glucose uptake rate was investigated through the calculation of flux control coefficients (FCCs), allowing comparison of the

distribution of metabolic control between the pure enzyme-constrained and hybrid model. In both conditions the FCCs obtained for hexokinases by the ecModel (YLR446W for respiration and HXK1 in fermentation) showed a value equal to 1, the highest value in their respective distributions, indicating that the overall glucose uptake rate is mostly governed by the activity of this enzymatic reaction step. In contrast, the constraints applied by the hybrid model distribute the control over the glucose uptake flux in a more even way across different enzymes and pathways, yielding FCCs of 0 for the different HXK isoforms in both metabolic regimes.

As a general trend, more FCCs with a high value (FCC>0.05) are obtained for fermentative conditions than for respiration, despite the use of the ecModel or hybrid model (Fig 4 and S3 Data). In the respiratory condition, the highest FCCs are concentrated in the oxidative phosphorylation pathway as well as around the branching point of pyruvate and PFK in glycolysis, whose activity is related to the connections between glycolysis and PP pathway. Moreover, the absence of glucose uptake control by lower glycolytic enzymes, and the prevalence of a non-zero FCC for PFK in both the ecModel and the hybrid model agrees with experimental evidence for mouse cell-lines in respiratory conditions [49]. For the fermentative condition, the highest FCCs are concentrated in the TCA cycle. Similarly to the respiration case, non-zero FCCs are present in the reaction steps surrounding the connecting points of different pathways, such as PFK, FBA, and TDH connecting glycolysis with the pentose phosphate pathway and reactions around pyruvate, which connect glycolysis with fermentation and the TCA cycle (Fig 4), this trend might indicate that in these branching points kinetic control is still a relevant mechanism governing fluxes.

### Deletion of the Snf1 in the hybrid model shows the importance of the Snf1 pathway in low glucose conditions and a connection between Snf1 regulation and chronological aging

To investigate how the individual signaling pathways contribute to changes in metabolic fluxes, the main component of each signaling pathway was deleted and flux changes were compared between the wild-type hybrid model and the knockout versions (S4 Data).

The Snf1 deletion was the only deletion showing any effect on the net fluxes in the respiratory condition (S4 Fig) while the Reg1, PKA and TOR deletions showed effects in fermentation conditions, consistent with the deletion experiments done with the Boolean model. The different mutants in fermentation do not induce major changes in net fluxes, however, the enzyme usage profile differs across the different mutants. Notably, the largest changes in terms of futile fluxes were observed in the TPI reaction, repressed in respiration by the Snf1 pathway and activated in fermentative conditions by either the PKA or Reg1 pathways. In respiration, Snf1 is also responsible for the futile fluxes through GPM, PGI and reduces the futile fluxes through FUM, MDH, PGM, and GAL10. The model simulations show a less diverse use of isoenzymes in all knockouts, which is most likely due to the reduction in the complexity of the regulatory layer. Considering the inherent property of flux balance analysis, any reduction in the regulatory network will be closer to the optimal distribution in which just the most efficient isoforms are used.

The Snf1 deletion exhibits an overall decrease in the flux towards respiration and a large decrease in flux through PPP, showing also a relatively strong downregulation of enzymatic steps surrounding pyruvate. The most significant changes are observed in NDE and PCK that are turned off and ALD6 which is turned on, implying that the Snf1 pathway is responsible for changing the acetate production via ALD6 to acetate production via ALD2, resulting in increased production of cytosolic NADH to the expense of the NADPH, which is compensated by increasing the flux through the pentose phosphate pathway as well as the additional use of

NDE. The use of ACS1 is abolished in the Snf1 deletion but not in other deletions, in the same manner as NDE. ACS1 has been shown to be upregulated in long-lived cells exposed to caloric restriction [50]. We compared the expression in our deletion simulations to experimental data of differentially expressed genes having a positive effect on chronological lifespan [51]. Out of 17 differentially expressed genes covered by our model, 6 were significant at a p-value = 0.05. Of those, 5 were also expressed in our model (S4 Data). RPE1 and CIT3 were upregulated in calorically restricted conditions. CIT3 is upregulated in our model by the Snf1 pathway while RPE1 is regulated by two mechanisms, one visible as a differential expression in the regulated WT compared to the ecModel, and one acting through the SNF1 pathway. RPE1 is also upregulated in the long-lived ade4 mutant strain together with the ADH5 gene. In our model ADH5 is not expressed in the ecModel, the hybrid model, or any of the simulated mutants. Cells treated with concentrates of media from cells grown under caloric restriction show an upregulation of PFK2. This is also shown in cells grown with the drug isonicotinamide (INAM) in which also PGK1 and ENO2 are upregulated. In the hybrid model, PGK1 and ENO2 are upregulated compared to the ecModel and none of the mutant strains showed any differential expression from the hybrid model of the WT. PFK2 similarly to RPE1 shows an upregulation by two mechanisms, one where the gene is upregulated in the hybrid model compared to the ecModel, and one where the Snf1 deletion shows a differential expression compared to the hybrid WT. The specific matrix of the gene regulatory network indicates that the mechanism is not related to a specific pathway. The regulation, of PFK2 and RPE1, is generated through a general regulatory effect caused by the network and not by regulation specifically acting on the gene itself.

## Discussion

The effects of nutrient-induced signaling on metabolism play an important role in maintaining organismal homeostasis and consequently understanding human disease and aging. To gain a better understanding of the interconnectivity between nutrient signaling and metabolism, we have developed a hybrid model by combining a Boolean and an enzyme-constrained model of metabolism, using a regulatory network as a link. More specifically, we have implemented a Boolean signaling network that is responsive to glucose and nitrogen levels and an ecModel of yeast's central carbon metabolism. The proposed framework has been validated using available experimental data resulting in an increased predictive power on individual protein abundances in comparison to individual models alone. Further, we were able to characterize the cells' deviation from optimal protein allocation and flux distribution profiles. The model is capable of reproducing the regulatory effects that are associated with the Crabtree effect and glucose repression. In respiratory conditions, the model showed regulation of genes known to be differentially expressed in long-lived cells. This regulation was shown by the hybrid model to act via both Snf1 dependent and independent mechanisms. In addition, the model showed that during fermentation, enzyme utilization is the more important factor governing protein allocation, while in low glucose conditions robustness and control are prioritized.

The integration of regulatory constraints is resulting in a highly constrained hybrid model. The downside of this approach is connected to the lack of information regarding the regulatory effects of transcription factors activation. In this work we assume a uniform proportional action for all gene targets, together with the other constraints of the model, resulting in a rather low effect on the regulatory action. Despite this, the hybrid model shows improved predictive power for individual enzyme demands and can qualitatively reproduce regulatory effects associated with glucose repression in fermentation conditions, suggesting that with this framework we can gain novel insight into the interplay between signaling pathways and metabolism.

Another limitation is the inclusion of only the central carbon metabolism, a potential extension of this work would include the addition of other pathways responsive to glucose signaling, like glycerol metabolism and fatty acid synthesis, enabling also the study of the regulatory effect on these pathways specifically with relatively few modifications in the hybrid model.

The current state-of-the-art methods for absolute quantification of protein abundance typically yield high experimental errors, spanning even over orders of magnitude, when measuring external standards with proteins of known concentration [52–54]. Such measurement errors are comparable to the average error in prediction of individual enzyme levels by the hybrid model. Further comparison of enzyme usage profiles against proteomics datasets revealed that, incorporation of a regulatory layer over an ecModel induces a diversified isoenzymes utilization profile, supported by experimental evidence, in contrast to a purely optimality-based approach (pure ecModel) in which this is rarely observed, especially in non-protein limited conditions (cellular respiration at low dilution rates).

The hybrid model shows that under regulation the NADH to support the electron transport chain is partly coming from the cytosol with the help of the mitochondrial external NADH dehydrogenase, NDE2. Overexpression of NDI1, in contrast to NDE1, causes apoptosis-like cell death which can be repressed by growth on glucose-limited media [47]. In our model regulation acts on both NDE and NDI which will lower the need for NDI1 expression and thus causing apoptosis-like cell death. The hybrid model gives the ability to determine that the Snf1 pathway alone is responsible for the shift to the additional use of NDE and NDI instead of only NDI. Snf1 is active in glucose-limited media and thus would help mitigate the phenotype of overexpressed NDI1. With our approach, we can attribute this effect to the Snf1 pathway specifically which a metabolic model alone would not be able to predict. Further, connecting Snf1 with the respiration-restricted apoptotic activity described previously [47], hybrid model contributes to the understanding of the role of Snf1 in chronological aging [50]. Additionally, the hybrid model could also predict the additional use of ACS1, not predicted by the ecModel or the SNF1 deletion, by increasing the flux through the ACS reaction. This phenotype of Snf1 has been indicated as an important factor in caloric restriction related extension of chronological lifespan in yeast [50]. When comparing differentially expressed genes in cells with extended chronological life span with genes affected by regulation in the hybrid model, both genes differentially expressed in caloric restriction conditions were regulated by the Snf1 pathway in the hybrid model, further strengthening the Snf1 mediated mechanism of extended chronological lifespan after caloric restriction. RPE1 and PFK2 were found in two different conditions leading to extended chronological lifespan and also showed two mechanisms of regulation in the hybrid model through systems biology effects, one general and one mechanism working through the SNF1 pathway. Interestingly all caloric restriction related conditions show at least one mechanism of the regulation working via the SNF1 pathway. This exemplifies how we can confirm known and possibly predict novel connections between signaling and metabolism when combined in a coherent framework.

Futile fluxes in the cell have been examined previously within the constraints of osmotics, thermodynamics, and enzyme utilization [55], where the osmotics are putting a ceiling on the allowed metabolite concentrations in the cell while thermodynamics govern the net fluxes through reactions. The induced futile fluxes can be explained by the fact that regulation included in the hybrid model will force the cell to use some enzymes even above its pathway flux requirements, adding robustness of metabolism to a constantly changing environment. The increase in flux in both forward and backward directions (i.e the increased futile flux through reactions) implies that these enzymes are working closer to their equilibrium and thus have a low flux control over the pathway flux, while enzymes with a strong forward flux have large flux control [56]. This feature is also displayed by our hybrid model, in which all

enzymatic steps with induced futile fluxes exert null control over glucose uptake (FCCs = 0). More enzymes in a pathway working close to their equilibrium results in robustness against perturbations as well as allow the pathway to be controlled and regulated through a few enzymes, however, this happens at the expense of inefficient utilization of enzymes as the cell needs to spend more resources to sustain a pool of enzymes that are carrying both forward and backward fluxes [55, 57]. Our predictions of several glycolytic steps forced to operate closer to their equilibrium by regulation (high futile fluxes induced for TDH, PGK, and ENO in respiration, and TPI, PGK, and GPM in fermentation) agree with experimental studies on *E. coli*, iBMK cells and *Clostridia cellulyticum*, which have suggested the utility of near-equilibrium glycolytic steps not just for providing robustness to environmental changes but also for enhancing metabolic energy yield [58].

Computation of FCCs showed that in respiration the glucose flux is tightly dependent on the activity of the enzymatic steps in oxidative phosphorylation, a high-energy yield pathway. In contrast, in the fermentative condition flux control is split between PFK, PYK, PDC, and several steps in the TCA cycle. Interestingly, the FCCs in the TCA cycle are decreased by around half, after applying the regulatory constraints in the hybrid model, providing hints of the importance of enhancing robustness in this pathway at high growth rates due to increased demand for biomass precursors. The prevalence of the highest FCCs in fermentation for PFK, PYK, and PDC (for both the ecModel and the hybrid model) indicates their important role as modulators of flux balance between glycolysis, PPP, and fermentative pathways at highly demanding conditions, suggesting that when entering fermentation, the cell sacrifices robustness to favor efficient enzyme utilization.

Comparison of enzyme usage and flux distributions between models and across conditions reveals that the effects of regulation are generally stronger for the respiratory condition, causing the arisen of more and higher futile fluxes; turning on reaction steps that are not required by optimal metabolic allocation (purely ecModel) and inducing higher upregulation of fluxes. These findings suggest that metabolic phenotypes are majorly shaped by regulatory constraints in low glucose conditions, whilst enzymatic constraints play a major role when glucose is not the limiting resource.

It was also found that the regulatory layer diminishes the strong flux control that hexokinase isoforms have over glucose consumption in both low and high glucose conditions to 0. The hexokinases in yeast, specially HXK2, have a central role in glucose signaling. It works both as an effector in the Snf1 pathway and also actively participates in the repression complex together with Mig1 in glucose repression during high glucose conditions [59]. Intuitively, it would be practical if an enzyme having these central and diverse tasks in the cell would not have such a high FCC as can be seen with the ecModel. When small perturbations in enzyme activity or concentration have large effects on glucose consumptions, allocating this enzyme to other parts of the cell such as the nucleus, participating in the repression complex, would be energetically expensive. Given the central role of hexokinase in glucose signaling, this would be of interest for further investigation and future studies.

Overall, in this work, we have shown how the hybrid modeling framework integrating nutrient-sensing pathways and central carbon metabolism can not only improve individual model predictions but can also elucidate how single components in the dynamic signaling layer affect metabolism at steady-state. We tested our model against both respiring and fermenting conditions and could not only predict known phenomena but also find novel connections. This methodology can be used to connect both original and readily available models in yeast to look at the interactions between signaling and metabolism. This can be applied to genome-scale and on different subsystems of metabolism and for different signaling systems (e.g. macronutrients or osmotic stress sensing). The availability of genome-scale models for

different organisms is constantly growing and with our increasing understanding of signaling systems and regulatory networks, the methodology developed in the course of this work can be adapted to many other organisms. Hybrid models, like the one proposed here, also provide a framework for hypothesis testing, as we demonstrated by knocking out several components of the nutrient-induced signaling network. In summary, we developed a methodology to investigate intrinsically different systems, such as signaling and metabolism, integrated into the same model, gaining insight into how the interplay between them can have non-trivial effects.

## Materials and methods

### Boolean model of nutrient-induced signaling pathways

Based on an extensive literature review, a detailed topology of the nutrient-induced signaling pathways TORC1, SNF1 and PKA accounting also for their crosstalks was derived and formalized as a Boolean network model using a vector-based modelling approach [15] **TORC1**: [8, 60–74]; **SNF1**: [75–101]; **PKA:** [7, 8, 12, 38, 102–115]; **crosstalks**: [36, 38, 40, 109, 116, 117].

The model consists of four different components: metabolites, target genes, regulated enzymes, and proteins. For the regulated enzymes, presence and phosphorylation state were considered whereas metabolites and target genes were only described by a single binary value indicating their presence and transcriptional state respectively. The state vectors were translated into a single binary value indicating the components' activity, allowing a better graphical depiction. In total, the model comprises 5 metabolites, 10 groups of target genes, 6 enzymes whose activity is altered upon nutrient signaling, and 46 proteins belonging to PKA/cAMP, the SNF1, and the TORC1 pathway, for detailed description, see S1 Text and S1 Table.

The availability of glucose and nitrogen was used as an input to the model and is implemented as one vector of binary values for each nutrient. This input enables to simulate the induction of signaling under different nutrient conditions, for instance, the addition of glucose and nitrogen to starved cells is represented by the vector 0|1 for both nutrients respectively. Here, 0 represents the starved or low nutrient condition and 1 the nutrient-rich condition. Based on this input and the formulation of the Boolean rules, a cascade of state transitions is induced. The simulation was conducted using a synchronous updating scheme meaning that at each iteration, the state vectors are updated simultaneously. The algorithm stops if a Boolean steady state is reached at which no operation causes a change in the state vectors. This process is repeated for each pair of glucose and nitrogen availabilities whereby the reached steady state for each nutrient condition serves as an initial condition for the next nutrient condition.

Since for many of the included processes, no information on the mechanisms causing reversibility was available, especially a lack in knowledge on phosphatases reverting phosphorylation was observed [15], gap-filling was conducted by including else-statements. This ensures that a component's state vector changes again e.g. if the conditions causing its phosphorylation are not fulfilled anymore. This gap-filling process guarantees the functionality of the Boolean model in both directions, meaning the simulation of state transitions occurring when nutrients (glucose and nitrogen) are added to nutrient-depleted cells as well as when cells are starved for the respective nutrients. Crosstalk mechanisms between the pathways were formulated as if-statements and can be switched off (0) or on (1). Furthermore, a simulation of knockouts of the pathways' components is possible by setting the value indicating their presence to 0.

### Enzyme-constrained metabolic model

A reduced stoichiometric model of *Saccharomyces cerevisiae's* central carbon and energy metabolism, including metabolites, reactions, genes, and their interactions accounting for

glycolysis, TCA cycle, oxidative phosphorylation, pentose phosphate, Leloir, and anaerobic excretion pathways, together with a representation of biomass formation, was taken as a network scaffold[29]. The metabolic model was further enhanced with enzyme constraints using the GECKO toolbox v1.3.5 [30], which considers enzymes as part of metabolic reactions, as they are occupied by metabolites for a given amount of time that is inversely proportional to the enzyme's turnover number ($k_{cat}$). Therefore, enzymes are incorporated as new "pseudo metabolites" and usage pseudo reactions are also introduced in order to represent their connection to a limited pool of protein mass available for metabolic enzymes. Moreover, all reversible reactions are split into two reactions with opposite directionalities in the ecModel, in order to account for the enzyme demands of backward fluxes. Several size metrics for the Boolean model, the metabolic network, and its enzyme-constrained version (ecModel) are shown in Table 1.

As the obtained ecModel has the same structure as any metabolic stoichiometric model, in which metabolites and reactions are connected by a stoichiometric matrix, the technique of flux balance analysis (FBA) can be used for quantitative prediction of intracellular reaction fluxes [118]. FBA assumes that the metabolic network operates on steady-state, i.e. no net accumulation of internal metabolites, due to the high turnover rate of metabolites when compared to cellular growth or environmental dynamics [119], therefore, by setting mass balances around each intracellular metabolite a homogenous system of linear equations is obtained. The second major assumption of FBA is that metabolic phenotypes are defined by underlying organizational principles, therefore an objective function is set as a linear combination of reaction fluxes which allows for obtaining a flux distribution by solving the following linear programming problem

$$\max : Z = C^T v$$

Subject to

$$S \cdot v = 0$$

$$lb \leq v \leq ub$$

**Table 1. Size metrics for the Boolean, original metabolic model, and its enzyme-constrained version.**

| Boolean model | |
| --- | --- |
| Metabolites | 5 |
| Target gene groups | 10 |
| Enzyme PTMs | 6 |
| Proteins | 46 |
| **Metabolic model** | |
| Reactions | 90 |
| Metabolites | 81 |
| Genes | 130 |
| Cellular compartments | 4 |
| **ecModel** | |
| Reactions | 324 |
| Metabolites | 111 |
| Enzymes | 127 |
| Promiscuous enzymes | 41 |
| Reactions with isoenzymes | 30 |
| Enzyme complexes | 11 |
| Reactions w/Kcat | 115 |

Where $C^T$, is a transposed vector of integer coefficients for each flux in the objective function ($Z$); $v$, is the vector of reaction fluxes; $S$, is a stoichiometric matrix, representing metabolites as rows and reactions as columns; $lb$ and $ub$ are vectors of lower and upper bounds, respectively, for the reaction fluxes in the system. Additionally, the incorporation of enzyme constraints enables the connection between reaction fluxes and enzyme demands, which are constrained by the aforementioned pool of metabolic enzymes

$$v_i = \sum_j k_{cat_{ij}} \cdot e_j$$

$$\sum_j^p Mw_j \cdot e_j \leq f \cdot \sigma \cdot P_{tot}$$

Where $k_{cat_{ij}}$ is the turnover number of the enzyme $j$ for the i-th reaction, as in some cases several enzymes can catalyze the same reaction (isoenzymes); $e_j$, is the usage rate for the enzyme $j$ in mmol/gDw h$^{-1}$; $Mw_j$, represents the molecular weight of the enzyme $j$, in mmol/g; $P_{tot}$, is the total protein content in a yeast cell, corresponding to a value of 0.46 g$_{prot}$/gDw [120]; $f$, is the fraction of the total cell proteome that is accounted for in our ecModel, 0.1732 when using the integrated dataset for *S. cerevisiae* in paxDB as a reference [121]; and $\sigma$ being an average saturation factor for all enzymes in the model.

This simple modeling formalism enables the incorporation of complex enzyme-reaction relations into the ecModel due to its matrix formulation, such as isoenzymes, which are different enzymes able to catalyze the same reaction; promiscuous enzymes, enzymes that can catalyze more than one reaction; and enzyme complexes, several enzyme subunits all needed to catalyze a given reaction.

### ecModel curation

As the ecModel was generated by the automated pipeline of the GECKO toolbox, several of its components were curated to achieve predictions that are in agreement with experimental data at different dilution rates. Data on exchange reaction fluxes at increasing dilution rates, spanning both respiration and fermentative metabolic regimes [43] was used as a comparison basis. Additionally, all unused genes in the original metabolic network were removed and gene rules for lactose and galactose metabolism were corrected according to manually curated entries for *S. cerevisiae* available at the Swiss-Prot database [122]. Gene rules and metabolites stoichiometries (P/O ratio) in the oxidative phosphorylation pathway were also corrected according to the consensus genome-scale network reconstruction, Yeast8 [21].

The average saturation factor for the enzymes in the model was fitted to a value of 0.48, which allows the prediction of the experimental critical dilution rate (i.e. the onset of fermentative metabolism) at 0.285 h$^{-1}$. ATP requirements for biomass production were fitted by minimization of the median relative error in the prediction of exchange fluxes for glucose, oxygen, $CO_2$ and ethanol across dilution rates (0–0.4 h$^{-1}$), resulting in a linear relation depending on biomass formation from 18 to 25 mmol per gDw for respiratory conditions and from 25 to 30 mmol per gDw for the fermentative regime.

### Hybrid model

A hybrid model consists of the Boolean model connected with the ecModel through a transcriptional layer that regulates its constraints on protein allocation (Fig 1). The active transcription factors act on the upper or lower bounds of the enzyme usage pseudo reaction depending on down- or up- regulation, respectively. The magnitude of the induced

perturbations is calculated according to previously calculated enzyme usage variability ranges, subject to a given growth rate and optimal glucose rate, expressed as

Upregulation:

$$lb^{reg}_{e_i} = e^{opt}_i + RF * (e^{max}_i - e^{min}_i)$$

Downregulation:

$$ub^{reg}_{e_i} = e^{opt}_i - RF * (e^{max}_i - e^{min}_i)$$

Where $lb^{reg}_{e_i}$ and $ub^{reg}_{e_i}$ represent the lower and upper bounds for the usage pseudo reaction of enzyme $i$ in the regulated model; $e^{opt}_i$, is a parsimonious usage for enzyme $i$ for a given growth and glucose uptake rates; $RF$, corresponds to a regulation factor between 0 and 1; $e^{max}_i$ and $e^{min}_i$ are the maximum and minimum allowable usages for enzyme $i$ under the specified conditions.

A distribution of parsimonious enzyme usages is obtained by applying the rationale of the parsimonious FBA technique [123], which explicitly minimizes the total protein burden that sustains a given metabolic state (i.e. fixed growth and nutrient uptake rates).

To connect the transcription factor activity with gene regulation we extracted regulation information from YEASTRACT and set a regulation level of 5% of the enzyme usage variability range for the simulations. When several transcription factors affect the same gene, the effects are summed up and the resulting sum is used as a basis for constraint. For example, if a gene is downregulated by two transcription factors (-2) and upregulated by one transcription factor (+1), the net sum would be (-1), thus the gene will be downregulated. In our model, an absolute sum higher than 1 will not cause a stronger regulation, as this additive process is just implemented to define the directionality of a regulatory effect.

## 2.5 Enzyme usage variability analysis

As metabolic networks are highly redundant and interconnected, the use of purely stoichiometric constraints usually leads to an underdetermined system with infinite solutions [124], in a typical FBA problem it is common that even for an optimal value of the objective function, several reactions in the network can take any value within a "feasible" range, such ranges can be explored by flux variability analysis [24].

In this study, enzyme usage variability ranges for all of the individual enzymes are calculated by fixing a minimal glucose uptake flux, for a given fixed dilution rate, and then running sequential maximization and minimization for each enzyme usage pseudo reaction.

$$enzyme\ usage\ variability\ range = e^{max}_i - e^{min}_i$$

Subject to:

$$v_{Glc_{IN}} = lb_{Glc_{IN}} = ub_{Glc_{IN}} = v_{Glc_{IN}}$$

$$v_{bio} = lb_{bio} = ub_{bio} = D_{rate}$$

This approach allows the identification of enzymes that are either tightly constrained, highly variable, or even not usable at optimal levels of biomass yield.

## Simulations

Cellular growth on chemostat conditions using minimal media with glucose as a carbon source, at varying dilution rates from 0 to 0.4 h$^{-1}$, was simulated with the multiscale model by the following sequence of steps:

1. Initially, the desired dilution rate is set as both lower and upper bounds for the growth pseudo reaction and the glucose uptake rate is minimized, assuming that cells maximize biomass production yield when glucose is limited [125, 126]

$$\min : \ v_{Glc_{IN}}$$

Subject to

$$D_{rate} \leq v_{bio} \leq D_{rate}$$

2. The obtained optimal uptake rate ($v_{Glc_{IN}}^{min}$) is then used as a basis to estimate a range of uptake flux to further constrain the ecModel.

$$v_{Glc_{IN}}^{min} \leq v_{Glc_{IN}} \leq (1 + SF) * v_{Glc_{IN}}^{min}$$

As $v_{Glc_{IN}}^{min}$ represents the minimum uptake rate allowed by the stoichiometric and enzymatic constraints of the metabolic network, possible deviations from optimal behavior may be induced by regulatory circuits. To allow the Boolean model to reallocate enzyme levels a suboptimality factor (*SF*) of 15% was used to set an upper bound for $v_{Glc_{IN}}$.

3. The ecModel is connected to the glucose-sensing Boolean model through the glucose uptake rate. At the critical dilution rate, the glucose uptake rate obtained by the ecModel is 3.2914 mmol/gDw h, this value is used as a threshold to define a "low" or "high" glucose level input in the Boolean model, represented as 0 and 1, respectively. For each dilution rate, the initial value of $v_{Glc_{IN}}^{min}$ is calculated and fed to the regulatory network, which runs a series of synchronous update steps until a steady-state is reached.

4. At steady state, the regulatory network indicates the enzyme usages that should be up and downregulated, for which new usage bounds are set as described above.

5. A final FBA simulation is run by minimizing the glucose uptake rate, subject to a fixed dilution rate, and the newly regulated enzyme usage bounds.

Gene deletions can also be set in the Boolean module and will result in activation or inactivation of transcription factors which then affect the constraints on the FBA model. We ran four simulations of deletion strains as follows: TOR1 and TOR2 (TOR deletion), Snf1 (SNF1 deletion), Tpk1, Tpk2, and Tpk3 (PKA deletion), and Reg1(Reg1 deletion).

## Proteomics analysis

Protein abundance data on respiratory and fermentative conditions were compared to protein usage predictions by the hybrid model to assess its performance. For the respiration phase, absolute protein abundances were taken from a study of yeast growing under glucose-limited chemostat conditions at 30˚C on minimal mineral medium with a dilution rate of 0.1 h$^{-1}$ [44].

For the fermentation phase, a proteomics dataset was taken from a batch culture using minimal media with 2% glucose and harvested at an optical density (OD) of 0.6 [45]. The dataset

given as relative abundances was then rescaled to relative protein abundances in the whole-cell according to integrated data available for *S. cerevisiae* in PaxDB [127], and finally converted to absolute units of mmol/gDw using the "total protein approach" [128].

We used three metrics for comparing the simulations with the proteomics data, the Pearson correlation coefficient (PCC), two-sample Kolmogorov-Smirnov (KS) test, and the mean of the absolute $\log_{10}$-transformed ratios between predicted and measured values (r). The PCC and the significance of the PCC were determined by a permutation test of n = 2000. The pathway enrichments were done using YeastMine [129] with the Holm-Bonferroni test correction and a max p-value of 0.05.

## Flux control coefficients

To investigate the relationship between enzyme activities and a given metabolic flux, control coefficients can be calculated for each enzyme in the model according to the definition given by metabolic control analysis (MCA) [56]:

$$FCC_{ij} = \frac{a_i}{v_j} \frac{\partial v_j}{\partial a_i}$$

In which $a_i = k_{cat_{ij}} e_i$ represents the activity of the *i-th* enzyme and $v_j$ is the flux carried by the *j-th* reaction. These coefficients represent the sensitivity of a given metabolic flux to perturbations on enzyme activities, providing a quantitative measure on the control that each enzyme exerts on specific fluxes.

As ecModels include enzyme activities explicitly in their structure, flux control coefficients can be approximated by inducing small perturbations on individual enzyme usages:

$$FCC_{ij} \approx \frac{k_{cat_{ij}} e_i}{v_j} \frac{\Delta v_j}{\Delta(k_{cat_{ij}} e_i)}$$

In our hybrid model, perturbations on individual enzyme usages ($e_i$) are induced in relation to a parsimonious usage ($e_i^*$) which is compatible with a given set of constraints

$$FCC_{ij} = \frac{e_i^*}{v_j^*} \frac{\Delta v_j}{\Delta(e_i - e_i^*)}$$

Perturbations equivalent to 0.1% of the parsimonious usage are used for each enzyme. For those cases in which the previously applied constraints do not allow such modification in a given enzyme usage, their activity is then perturbed by operating on the corresponding turnover number for the enzyme-reaction pair ($k_{cat_{ij}}^* = 0.001 * k_{cat_{ij}}$) to simulate a perturbation in their overall activity.

## Supporting information

**S1 Text. Supporting information on the Boolean layer.** Includes a detailed description of mechanisms reflected in the Boolean model of nutrient signaling as well as open questions of dynamics and model gaps.
(DOCX)

**S2 Text. Supporting Information on the hybrid model.** Includes detailed information on the analysis of protein prediction and deletion strain simulations.
(DOCX)

**S1 Fig. Transition map of components in the Boolean model separated by their respective pathway where the blue color indicates activity.** Simulations are made with all crosstalk turned on. Panel (A) shows the simulation dynamics going from nutrient-depleted conditions to nutrient-rich conditions and panel (B) shows the simulation dynamics going from nutrient-rich conditions to nutrient depletion.
(PDF)

**S2 Fig. Steady-state map of components in the Boolean model when knock-out (KO) strains are simulated from wild type (WT) to KO where the blue color indicates activity.** Panel (A) shows the KO behavior in low nutrient conditions compared to the WT and panel (B) show the KO behavior in high nutrient conditions compared to the WT
(PDF)

**S3 Fig. Exchange fluxes for the hybrid model plotted over experimental data.** Simulations showed a median relative error of 9.82% in the whole range of dilution rates from 0 to 0.4 h-1.
(PDF)

**S4 Fig. The fluxes through the core reactions in the metabolism are represented by the width of the connectors where dotted lines represent zero flux.** The color of the connectors represents the change in flux from the wild type (WT) hybrid model compared to the SNF1 deletion hybrid model. The FCCs are represented in the model where the WT is compared to the SNF1 deletion case.
(PDF)

**S1 Table. Rules and references associated with any field of any of the Boolean vectors in the Boolean module.**
(DOCX)

**S2 Table. Summary of the statistics done comparing the ecModel and the hybrid model in their ability to predict protein abundance.**
(DOCX)

**S1 Data. Data related to enzyme usages and protein prediction.**
(XLSX)

**S2 Data. Data related to fluxes.**
(XLSX)

**S3 Data. Data related to FCC.**
(XLSX)

**S4 Data. Data related to mutant strain simulations.**
(XLSX)

## Acknowledgments

We would like to thank members of the Hohmann, Nielsen, and Cvijovic labs for valuable input. Special thanks to Avlant Nilsson for his contributions to the curation of the original metabolic network used in this study and valuable discussions on the role of enzyme constraints.

## Author Contributions

**Conceptualization:** Linnea Österberg, Marija Cvijovic.

**Data curation:** Linnea Österberg, Iván Domenzain, Julia Münch.

**Formal analysis:** Linnea Österberg, Iván Domenzain.

**Funding acquisition:** Jens Nielsen, Stefan Hohmann, Marija Cvijovic.

**Investigation:** Linnea Österberg, Iván Domenzain, Julia Münch.

**Methodology:** Linnea Österberg, Iván Domenzain, Julia Münch.

**Project administration:** Linnea Österberg, Marija Cvijovic.

**Resources:** Jens Nielsen, Marija Cvijovic.

**Software:** Linnea Österberg, Iván Domenzain.

**Supervision:** Jens Nielsen, Stefan Hohmann, Marija Cvijovic.

**Validation:** Linnea Österberg, Iván Domenzain.

**Visualization:** Linnea Österberg, Iván Domenzain, Julia Münch.

**Writing – original draft:** Linnea Österberg, Iván Domenzain.

**Writing – review & editing:** Jens Nielsen, Stefan Hohmann, Marija Cvijovic.

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
