## [Decision Letter · Decision Letter 0]

8 Dec 2020

Dear Dr. Cvijovic,

Thank you very much for submitting your manuscript "A novel yeast hybrid modeling framework integrating Boolean and enzyme-constrained networks enables exploration of the interplay between signaling and metabolism" for consideration at PLOS Computational Biology.

As with all papers reviewed by the journal, your manuscript was reviewed by members of the editorial board and by several independent reviewers. In light of the reviews (below this email), we would like to invite the resubmission of a significantly-revised version that takes into account the reviewers' comments.

We cannot make any decision about publication until we have seen the revised manuscript and your response to the reviewers' comments. Your revised manuscript is also likely to be sent to reviewers for further evaluation.

Sincerely,

Attila Csikász-Nagy

Associate Editor

PLOS Computational Biology

Jason Papin

Editor-in-Chief

PLOS Computational Biology

Reviewer's Responses to Questions

**Comments to the Authors:**

Reviewer #1: In this work, the authors present a hybrid yeast model combining signaling, regulation and metabolism. The model offers advantages over purely metabolic models by allowing to analyse the effects of nutrient sensing. The final model is a combination of a signaling and regulatory network for carbon and nitrogen sensing, manually curated by the authors, with a core metabolic model of yeast metabolism obtained by reducing the network of the yeast gecko model. Overall the paper presents a methodological advancement in the combination of boolean-based signaling model with a protein-constrained metabolic model. While the scientific aspect of the manuscript seems solid, and of general interest, for me the main issue is the writing and structure of the manuscript. I found the text not very well written and difficult to follow. I believe the manuscript would benefit from substantial rewriting. Please see further details bellow.

Major issues:

* In the introduction the reader is only pointed to a diagram that doesn’t explain much of the simulation method. The details of simulation are explained at the end, but a brief description (pointing to the methods section for a more detailed description) should be given before explaining the main results.

* Each subsection of the results immediately begins with an enumeration of results. A bit more context into the motivation for each part, and an explanation of why these particular simulations were selected with respect to that motivation, would better help the reader to interpret the results.

* I don’t understand why are there only 3 figures in the main text and all the others go into supplementary material. Some of these figures are essential for understanding the method and the main findings in the paper.

* The supplementary file itself is very confusing. There are full paragraphs of text mixed with figures and tables. This file needs to be better organized into supplementary figures, supplementary tables and supplementary text. Also, a lot of the supplementary text contains interpretation of results and discussion. There is no explanation given to the reader at any point on why some of the text is moved into supplementary and what one should expect to find there.

Minor issues:

* Please explain how the thresholds are decided for converting the glucose and nitrogen concentrations into binary values.

* Please specify the growth medium used during simulation.

* In lines 231-239 there are multiple p-values presented. Please mention also the statistical test used, sample sizes, effect size, and the respective test statistic.

* Fig 3A looks very suspicious, some values reach almost 10^-10, which is probably below the solver precision. Log-changes are a very strange way to compare experimental and simulated fluxes. Why not use the mean squared error (MSE) instead ?

* Fig 3B: why showing only these 3 proteins when there are more proteomics data ?

* Fig 3D: the net flux for PGK and GPM seems higher than futile flux, how is this possible?

Reviewer #2: This paper aims do provide a novel hybrid modelling framework to integrate (nutrient-induced) signalling, regulation and metabolism, which is applied to yeast and used to perform some simulations.

The overall idea for the framework is well designed and the goal is worthy of research, being a major aim of systems biology. However, I believe that the work has several limitations, mainly regarding the results presented, which do not seem to justify the authors' claims.

Firstly, regarding the model, it should be clear from the beginning this not a genome-scale model, even in the metabolic part and that the signalling and regulatory layers are very limited. For instance, this is not clear in the abstract of the work.

Also, the reported simulations are very limited and provide results that are not convincing. The errors provided for the proteomics datasets are all in the high orders of magnitude (2 to 4); while improvements are visible the obtained errors are still around 100 fold. Also, the reported use of iso-enzymes is only somewhat visible in fermentation.

Finally, the Crabtree effect reported was already reported for GECKO models alone, so it is not clear where the improvement is. The relationship to ageing is also not convincing and its significance not well explained.

So, although I value the work on the framework and the thorough work on the signalling part of the model, the overall result does not seem to justify the publication.

There are also some language problems; the overall paper should be read, as there are a few typos (e.g. investigat - 28; singaling - 39, trough - 548) and in many cases sentences are confusing missing commas to make them more readable (as an example consider the sentence in rows 89-90). The language in S1 text is even less carefully written (consider one example of a sentence: "As can be seen by the summery statistics, prediction of individual proteins are difficult. ").

**Have all data underlying the figures and results presented in the manuscript been provided?**

Reviewer #1: None

Reviewer #2: Yes

PLOS authors have the option to publish the peer review history of their article (what does this mean?). If published, this will include your full peer review and any attached files.

Reviewer #1: No

Reviewer #2: No
---

## [Decision Letter · Decision Letter 1]

18 Mar 2021

Dear Dr. Cvijovic,

We are pleased to inform you that your manuscript 'A novel yeast hybrid modeling framework integrating Boolean and enzyme-constrained networks enables exploration of the interplay between signaling and metabolism' has been provisionally accepted for publication in PLOS Computational Biology.

Best regards,

Attila Csikász-Nagy

Associate Editor

PLOS Computational Biology

Jason Papin

Editor-in-Chief

PLOS Computational Biology

Reviewer's Responses to Questions

**Comments to the Authors:**

Reviewer #2: The authors successfully addressed by main comments in the revised version.

I still believe the language issues are not completely solved (it has improved) and a re-read of the manuscript could help.

**Have all data underlying the figures and results presented in the manuscript been provided?**

Reviewer #2: Yes

PLOS authors have the option to publish the peer review history of their article (what does this mean?). If published, this will include your full peer review and any attached files.

Reviewer #2: No

---

## [Editor Report · Acceptance letter]

30 Mar 2021

PCOMPBIOL-D-20-01648R1 

A novel yeast hybrid modeling framework integrating Boolean and enzyme-constrained networks enables exploration of the interplay between signaling and metabolism

Dear Dr Cvijovic,

I am pleased to inform you that your manuscript has been formally accepted for publication in PLOS Computational Biology. Your manuscript is now with our production department and you will be notified of the publication date in due course.

With kind regards,

Alice Ellingham
